# Healthcare Professionals’ Perspectives on HPV Recommendations: Themes of Interest to Different Population Groups and Strategies for Approaching Them

**DOI:** 10.3390/vaccines12070748

**Published:** 2024-07-06

**Authors:** Carlos Murciano-Gamborino, Javier Diez-Domingo, Jaime Fons-Martinez

**Affiliations:** 1The Foundation for the Promotion of Health and Biomedical Research of Valencia Region (FISABIO), 46020 Valencia, Spain; javier.diez@fisabio.es; 2CIBER de Epidemiología y Salud Pública, Instituto de Salud Carlos III, 28029 Madrid, Spain

**Keywords:** HPV vaccination, DEI, communication

## Abstract

As a flagship of the Europe’s Beating Cancer Plan, the European Commission supports EU member states’ efforts to strengthen and expand the routine vaccination of girls and boys against human papillomavirus (HPV). Populations across Europe have grown in diversity, and health systems must adapt to meet the specific needs of increasing diversity. Healthcare professionals (HCPs) must strive to communicate HPV vaccine information in a culturally sensitive manner and address specific concerns related to cultural beliefs, trust in health systems and perceived risks. The objectives of this exploratory study are to identify which themes are most frequently raised during the recommendation of vaccination to minors based on the characteristics of the population (religion, region of origin, gender, level of education and language proficiency) and to collect strategies to improve communication with a diverse population. A survey was distributed through various European public health institutions to HCPs in the region and their networks. The survey included multi-response questions (themes addressed during vaccination recommendation based on population characteristics) and open-ended questions (own qualitative comments and strategies). The most common issues that arise during vaccine recommendation are a lack of knowledge, followed by misinformation. Differences were detected according to the population characteristics. Suggested strategies to improve HPV vaccine recommendation focused on the following aspects: affordability; sexuality and gender; communication platforms; multilingualism; quality of care; school collaboration. HCPs report differences according to the characteristics of the population receiving the recommendation. Personalisation of the recommendations would help to optimise the decision-making process for some groups.

## 1. Introduction

The European Commission considers it a public health priority to combat preventable causes of cancers, as reflected in the Europe’s Beating Cancer Plan. As a flagship initiative of this plan, the Commission supports the efforts of EU member states in strengthening and expanding the routine vaccination of girls and boys against human papillomavirus (HPV) to eliminate cancers caused by HPV. The goal is that by 2030, at least 90% of the EU target population of girls will be fully vaccinated and that there will be a significant increase in the vaccination of boys [1]. This plan reflects and enhances the trend in EU member states of including HPV vaccination for both girls and boys in their vaccination schedules [2]. On the other hand, Europe is experiencing a significant increase in diversity due to the presence of more than ninety million international migrants in the World Health Organization (WHO) European Region [3]. This influx of diverse populations brings forth new challenges for healthcare systems and professionals in effectively addressing the needs of these individuals. The increasing multiculturalism of societies makes it necessary for healthcare personnel to adopt culturally sensitive care. This adaptation improves both the understanding of the patients’ needs and behaviours and communication with them. Intercultural communication offers a solution to bridge the gap between healthcare providers and diverse patient populations. By embracing intercultural communication strategies, healthcare professionals (HCPs) can effectively convey important information about healthcare, such as the HPV vaccine (HPVV), in a manner that is culturally appropriate and easily understood by patients from diverse backgrounds [4]. One critical aspect of intercultural communication in healthcare is addressing and overcoming any prejudices that HCPs may hold. Prejudices can negatively impact the quality of care and patient outcomes, particularly when dealing with diverse populations [5]. HCPs must be aware of their own biases and actively work to avoid them, ensuring that their communication is free from discrimination and cultural insensitivity, as well as facilitating strategies for the integration of the diverse population. By fostering an open and unbiased approach, HCPs can build trust and establish effective relationships with their patients [6].

Moreover, a study by Fernández de Casadevante and colleagues in Denmark points out that communication between HCPs and the immigrant population is different compared to communication with the majority ethnic population, often leading to poor mutual understanding and reduced compliance [7]. Cross-cultural communication in primary care is often difficult, leading to unsatisfactory, substandard care: training primary care teams results in a more tolerant attitude and more effective communication, with better focus on patients’ needs [8].

In the context of the HPVV, understanding patients’ cultural practices and beliefs is crucial for effective communication. Therefore, HCPs should strive to communicate vaccine information in a culturally sensitive manner and address specific concerns related to cultural beliefs, trust in healthcare systems and perceived risks, taking into account that cultural practices are not merely individual preferences but are determined by an individual’s position within the social structure [9]. By recognising the interplay between individual choices and structures (social context), HCPs can tailor their communication to address specific concerns and overcome barriers to vaccination.

### Rationale for Study Dimensions

Religion, culture, gender and socio-economic factors have been identified as some of the main influencers for vaccine hesitancy [10], and patient’s characteristics, such as gender, education or ethnicity, have been also identified as the main factors that influence communication [11]. In the case of the recommendation of the HPV vaccination, their influence may be higher than for other vaccines because HPV is a sexually transmitted infection. For the purpose of this study, these variables have been translated into the following main study dimensions: religion, region of origin/ethnicity, gender, educational level and language proficiency. Taking this into account and considering the diversity within the EU, it is important to explore the impact of these factors on the perception and acceptance of the HPV vaccine. They can influence individuals’ health beliefs, attitudes towards vaccination and interaction with healthcare systems and information sources. Understanding these influences helps in formulating targeted strategies to improve communication and vaccination rates among diverse populations.

## 2. Materials and Methods

The findings presented in this article emerge from a larger research study carried out in the European project PROTECT-EUROPE (EU4H-2021-PJ2, Project GA No. 10108004) to increase HPV vaccination rates.

### 2.1. Aims of the Study

Although cultural differences in the perception of health and disease have been widely addressed by various authors from different disciplines, as well as strategies to improve intercultural communication, little literature has been developed in Europe on the act of recommending HPV vaccination and on the perception of the HPVV according to the characteristics of adolescents and their parents/guardians (APGs).

The objectives of this exploratory study are as follows:To identify the themes that tend to emerge in conversations between HCPs and APGs during HPV vaccination recommendation according to the characteristics of the population served (religion, region of origin, gender, educational level and language proficiency).Identify strategies to approach HPVV recommendations taking into account the characteristics of the population.

Given the scarcity of studies in Europe that take into account as many different dimensions as the current one, we expect that this exploratory study will allow for the identification of trends among the population that can guide HCPs in making their recommendations and the compilation of some strategies that could contribute to improving the acceptance of the HPVV in some population groups and serve as a basis for future hypotheses and studies.

### 2.2. Tools

The study was carried out by means of an exploratory survey (see Appendix A). The survey was designed and conducted using the REDCap^®^ platform, distributed by email and self-completed by the participants. The survey is divided into two parts:Themes that emerge in conversation with APGs in recommending the HPVV according to the characteristics of the population served. It includes closed multi-response questions and qualitative questions so that the respondent can add other themes to those proposed.Strategies proposed to improve communication about and recommendation of the vaccine. Open-ended qualitative questions.

In this survey, HCPs identified the population they served based on the following dimensions: religion; region of origin of immigrants and descendants of immigrants, ethnicity and/or legal status in the country; gender (of the young adolescent eligible for vaccination); educational level and language proficiency. Based on their experience, the respondents identified themes that tended to appear in conversations with the population with a certain characteristic (Survey Part A). The common themes during the HCP-APG communication about HPV and its vaccine were obtained from the categorisation of the main barriers and facilitators to vaccination identified in the systematic review conducted by Zheng et al. in 2021 [12], resulting in the following:Taboos related to people’s sexuality that may affect the conversation between a healthcare professional and an adolescent or his/her parents/guardians.Negative perceptions of the HPVV in their environment (e.g., that it encourages promiscuity) and how this perception influences their decision to be vaccinated (or not).Considerations about the lack of benefit of the vaccine at the time, HPV being a sexually transmitted infection and given that it is administered at a very young age.Misinformation about the HPVV, such as lack of efficacy and safety or unproven adverse effects (e.g., that it causes infertility).Lack of knowledge about HPV infection and its consequences.When offering the vaccine to a male, the false belief that it only has health benefits for men who have sex with men (this barrier was not extracted from the systematic review conducted by Zheng et al.; it was included by the authors due to increasing recommendations for a gender-neutral vaccination among European countries).Difficulties in accessing the health system and completing the recommended schedules (two or three doses).

In addition to these options, a question was added at the end of each section to allow the participants to add their own qualitative comments on that characteristic by asking, “Do you think there are any issues that come up recurrently in conversation with people of any [characteristic, e.g., religion] that have not been mentioned before? Please describe the topic/issue and the population you think it affects”.

In the second part of the survey (Part B), through open-ended questions, respondents were asked for recommendations to improve HPV- and HPVV-related communication based on the characteristics of the population served.

### 2.3. Sampling

The survey was distributed through different European public health institutions among professionals in the region and their networks following snowball sampling. The survey targeted healthcare professionals who were in direct contact with patients, practising in Europe and who provided information on and/or offered adolescents the HPV vaccine; there were no defined exclusion criteria.

As this is an exploratory study, the aim is not to obtain a sample size and statistical significance but to obtain a broad view of the phenomenon studied [13]. For this reason, the sample size objective was not fixed.

### 2.4. Analyses

Different analyses were carried out to analyse the survey data:1.Survey Part A: Themes
A frequency analysis of the seven themes identified in each population was performed.Based on the examples of thematic groupings provided by Pop [14] and Heffernan [9], the themes were grouped into 3 categories: (I) Moral/cultural themes (A, B and C), (II) Informational themes (D, E and F) and (III) Access-related themes (G). Based on these categories, a scatter plot of the different populations was made.2.Survey Part B: Recommendations
Concerning the information collected in the open-ended questions, the recommendations provided were analysed following a qualitative content methodology, creating categories and grouping the responses into them to generate grouped recommendations.


## 3. Results

### 3.1. Characteristics of the Sample

A total of 214 HCPs completed the survey, with a predominance of women, nurses and school medicine professionals practising in Croatia or Spain. Table 1 shows their demographic characteristics.

### 3.2. Characteristics of the Population Served

Regarding the description of the population that the respondents usually serve (Table 2), it was observed that most of the respondents served people of the Catholic religion (n = 200, 93.46%), followed by Muslims (n = 90, 42.06%). In terms of the origin of the patients, most of the respondents serve other European countries (n = 129, 60.28%), the Near or Middle East (N = 69, 32.24%), Latin America (n = 66, 30.84%) and North Africa (n = 64, 29.91%).

With respect to gender, the majority of respondents serve girls (n = 203, 94.96%) and boys (n = 195, 91.12%). Finally, in relation to the level of education and language skills of the served population, 57.48% (n = 125) of the respondents claimed to serve people with a low educational level, and 31.78% (n = 68) reported serving people with low language skills.

### 3.3. Survey Part A: Themes That Emerge in the Conversations According to the Characteristics of the Population Served

Once the population served by each participant in the survey was identified, they were asked to identify, from their point of view, which themes tended to come up in conversations with adolescents and their parents or guardians (APGs) when recommending the HPV vaccination to minors, depending on the characteristics of the population served.

Appendix A shows the themes that arise with each population. It should be borne in mind, on the one hand, that respondents could identify more than one theme for each population (multiple responses) and, on the other hand, that the percentages in the table reflect the number of respondents who said they served that population and who considered that this theme appeared in their conversation. Therefore, the denominator is the number of respondents who serve that population and not the total number of respondents.

Figure 1 shows the different populations according to the type of demand in the conversation about HPV. For its construction, the vertical axis represents the themes of an informational nature, and the horizontal axis the ones of a moral/cultural nature. Finally, the theme related to access to the health system has been represented by means of a colour scale (blue = minimum = ROA (0.0%); orange = maximum = GRT (28.6%)).

After standardising, four quadrants were defined as follows:QI: The percentage of HCPs reporting the occurrence of informational topics in conversations with these populations is higher than the average for all populations, but the percentage of HCPs reporting the occurrence of cultural/moral topics is lower than the average for all populations.QII: The percentage of HCPs reporting the occurrence of both informational and cultural/moral themes in conversations with these populations is higher than the average for all populations.QIII: The percentage of HCPs reporting the occurrence of cultural/moral themes in conversations with these populations is higher than the average for all populations, but the percentage of HCPs reporting the occurrence of informational themes is lower than the average for all populations.QIV: The percentage of HCPs who reported the occurrence of both informational and cultural/moral topics in conversations with these populations is lower than the average for all populations. The non-appearance of certain themes on a regular basis in conversations with certain population groups should not be interpreted unequivocally as a lack of interest in these topics but instead may be seen as a taboo or sensitive subject that they do not want or know how to share with an HCP. Even so, it should always be borne in mind that the information presented here is representative of trends and not categorical differences between populations so that all recommendations should be adapted according to the interlocutor and not because they present certain characteristics or others.

### 3.4. Survey Part A: What Themes Emerge According to the Characteristics of the HCP?

Interaction between HCPs and APGs not only depends on the characteristics of the latter but is also influenced by factors associated with the professional [11].

In the analysis of the results by the age group of health professionals (Table 3), variations are observed. As their age increases, the percentage of professionals who say they talk about these themes during the recommendation of vaccination decreases, except with the theme of access (G), where variations are minor, and negative perception of the environment (B), where there is a difference between the youngest group of professionals (67.39%) and the next (55%), but the positive trend recovers as the age group increases (56.6% in the group of 46–55 and 59.18% in the group of those over 56). On the other hand, with regard to the speciality of the professional (Table 3), a higher proportion of themes appear in the conversations of paediatricians and school medicine professionals than in those of other healthcare profiles.

### 3.5. Survey Part B: Considerations and Strategies Proposed by HCPs to Improve Communication about and Recommendation of HPV Vaccination

In the section “Proposals to improve one-to-one communication” of the survey (see Appendix A), the respondents were asked for recommendations on the issues raised. In most cases, the recommendations focused more on “things to consider” during general HPV-related and HPVV-related communication, focusing on different dimensions, than on recommendations on how to overcome barriers that may arise with specific populations.

The responses were grouped into the following categories (Table 4): affordability, school collaboration, informative materials and campaigns, multilingualism, sexuality and gender and quality of care.

## 4. Discussion

Communication between HCPs and APGs about vaccination is essential to reduce their doubts and concerns and to encourage their acceptance of vaccination. HCPs are sometimes faced with questions and situations for which they are not prepared, so it is important to be able to anticipate what types of issues may be of concern to their patients and to establish strategies to address them [15].

The results of this study, which are based on the experience of the HCPs, indicate the perception of differences in the frequency with which certain topics come up in HCP-APG conversations when advising the HPV vaccination. This is consistent with the systematic review by Fernández de Casadevante and colleagues of European studies, which noted that ethno-cultural and educational factors may be important when it comes to HPVV uptake [16].

These perceived differences make it advisable to adapt the discourse and discussion when recommending vaccination according to the characteristics of the population. The need for adaptation has been identified in the literature in the field of HPV vaccination, as well as with other vaccines or in other health settings [17,18,19]. For example, with respect to vaccines in general, Limaye et al. [15] identify tailoring information to increase the relevance of the message as an approach that can improve vaccine uptake, and with respect to other health domains, such as informed consent processes, the ‘Guidelines for Tailoring the Informed Consent Process in Clinical Studies’ identify co-creation, tailoring information to the needs and preferences of the population and inclusive information preparation as key ideas for developing informational materials [20].

### 4.1. Themes That Emerge in the Conversations According to the Characteristics of the Population Served

In terms of the results of our survey, there is a general need to improve the information that the population receives regarding HPV and HPV vaccination in order to combat both lack of knowledge (E) and misinformation (D), given that these two issues tend to be predominant in most groups. It is important to mention that lack of information or knowledge or the perception of not needing the vaccine are precisely some of the main barriers to vaccination identified by the literature [18]. In addition, false or misleading information, as well as information about the risks of vaccination, tends to be more widely disseminated on social media than messages in support of vaccination [21,22,23].

Regarding the consideration of sexuality issues as taboo (A) is common to many cultures and religions. Several studies indicate the difficulties parents have in talking to their children about sexuality [24,25,26] and even point to the desirability of a third party helping to overcome these intergenerational barriers [25]. This highlights first the role that HCPs can play in this respect; second, the convenience of offering the possibility of talking alone with adolescents or with parents or guardians and third, the difficulty of dealing with the subject when the person is not fluent in the language and requires a person from their family environment to be a translator [6].

With regard to the issue of access, difficulties in completing the vaccination schedule, access to health services and completing the recommended HPV vaccination schedule (G), the population from Africa, the population with an irregular legal status, the GRT population and those with poor proficiency in the language of the host country are those for whom these issues tend to appear during the recommendation. Given the recent generalisation of HPVV financing in all EU countries (Romania being the latest) [2], the problem of these groups becomes relevant not only because of economic barriers but also because of difficulties in accessing health services.

#### 4.1.1. Religion

Regarding the differences in the themes that emerge during the recommendation according to religion, the Protestant population appears to be the one in which moral/cultural themes emerge the least, while Catholics and Muslims are the ones with whom more themes of this nature are discussed. The perception that moral/cultural issues appear to a lesser extent with Protestants than with other groups contrasts with the results of Mollers et al., who indicate that the main reason for the Protestant population rejecting the vaccine is precisely the fact that HPV is considered a sexually transmitted infection [27]; this may be because they consider it a fundamental aspect and not subject to discussion, so it does not emerge in the conversation. In fact, the lower presence of these issues among the Protestant population should not be understood as a greater acceptance of the vaccine, since, as indicated by Mollers et al., there is a greater tendency towards acceptance of the vaccine among the Catholic population and a lower tendency among the Protestant population. Furthermore, these authors also indicate that anti-vaccine movements are more common in countries with a high proportion of the population professing the Protestant religion [27].

The study conducted by Hittson et al. among religious students, mostly Christians of various denominations in the United States, found that frequency of religious service attendance was positively correlated with a negative attitude towards HPV vaccination [28], and this is also noted by Brabin et al., who identify a lower acceptance of HPV vaccination among parents with strong religious or cultural views [29]; this could be a particularity of the HPVV itself, as Kibongani Volet et al. point out, based on studies in sub-Saharan Africa, that religious groups tend to have higher coverage of full immunisation of children than the rest of the population [30]. On the other hand, a study of young people in Italy found a later sexual debut among devout Christian youth compared to non-religious youth [31]. These studies would support the results of our survey regarding the prevalence of the emergence of the issue of a lack of benefit of the vaccine at this age among the Catholic population.

Regarding the position of religious authorities on the HPVV, in the case of Catholicism, the Vatican has not officially adopted a position, while some bishops have spoken out in favour of or against its use [32], and certain Catholic authorities such as the National Catholic Bioethics Centre [33] and the Catholic Medical Association [34] have made statements indicating that the use of this vaccine is morally acceptable in appropriate situations. This certain openness towards HPV vaccination, coupled with the process of secularisation [35] observed in the predominant Christian streams in many Western countries, would make moral and cultural issues depend on a variety of factors and not only on religious affiliation. These processes may partly explain why the negative perception of vaccination and its influence on the decision to vaccinate were the prominent cultural and moral themes in conversations about HPV vaccination with Catholics, as reported by a significant proportion of HCPs serving this population, exceeding the proportion observed for other religions.

It is important to note that although in many cases various religious authorities, including Catholic authorities, advocate celibacy as the best preventive option, some young people continue to engage in HPV risk behaviours, such as oral sex, believing that they do not infringe on celibacy, as they continue to remain “technical virgins” [28]. This reinforces the importance of both vaccination and information about the disease and its modes of transmission, supporting the results of our survey, where the percentage of HCPs serving this population indicates the emergence of themes related to misinformation about the HPVV (D) and a perceived lack of benefit (C) more frequently than in the population professing other religions.

With regard to the Muslim population, it is characterised by the highest proportion of HCPs who have indicated that themes of taboos (A), a lack of knowledge about the disease and its consequences (E), the benefit to children (C) and access to the vaccine (G) emerge. Hamdi notes that the Muslim population tends to be more conservative about sexual behaviour compared to other Western populations because they have more traditional religious and social norms, which pose a barrier to vaccination [36], and Muslims have lower rates of acceptance of vaccination in general according to the Report of the State of Vaccine Confidence in EU for 2022 [37].

However, in recent years, there have been profound changes in the way the genders and sexual life are conceived and related, and the traditional model no longer applies to younger generations as it did in the past. Thus, we understand that this awareness of the existence of a social change in the way of understanding and behaving in relation to sexuality may partly explain the results of our survey and that it is in this population that a greater proportion of HCPs have identified the emergence of taboo (A) and information issues (E), although the emergence of these issues should not be confused with a greater tendency to accept the vaccine, as Pratt et al. point out in reference to the Somali Muslim population living in the United States that they are still less likely to accept vaccination than the general population [38], and the emergence of taboo issues may also indicate that they do not want to discuss the issue or at least not in the presence of family members. In this regard, Salad et al. report that in their study of Somali Muslim mothers and daughters living in the Netherlands, they only included the mother or daughter from the same family, as sexuality and health issues are sensitive to discuss in the Somali community and could lead to discomfort [24]. Kouhen and Ghanmi also note the difficulty of discussing intimate issues even in a healthcare setting [39].

The fact that vaccination is a preventive activity and not a treatment may also be important given that, as Hamdi points out, in Muslim societies, people tend not to admit or reveal their sins [36], so this preventive component given at a young age (prior to potential sin) could facilitate addressing the theme during the recommendation of vaccination.

In terms of the results on the Hindu population, HCPs serving this population did not highlight any specific themes in their conversations with them, showing lower frequencies than the average for all religions in all topics except access (G), which is striking considering that previous studies highlight the low vaccination rates of this group, such as the study by Cofie et al. in which they identify women from the Indian subcontinent as the demographic of immigrant women in the United States with the lowest HPV vaccination rates [40]. On the other hand, Marlow and colleagues point out that sex-related issues appear as reasons for refusing vaccination more frequently in the Indian or Pakistani population than in the white British population [41]. However, this lower rate may not be due to intrinsically conversational issues but to other factors since, as Ratnasamy and Chagpar point out, the second-generation population in the USA does not feel as much stigma when talking about these themes [42]. This is particularly noticeable given that in the countries in our sample, immigration from India is more recent and therefore probably less integrated, so these issues would be expected to emerge more frequently in conversations with HCPs.

#### 4.1.2. Region of Origin

With respect to the region of origin of the population served, it is striking how this category evokes a greater perception than the rest of the categories of the appearance of themes related to taboos (A), with the average for region of origin being higher than that of the rest of the categories, while it is also the one that evokes the least appearance of the theme of negative perceptions of the HPVV by their environment (B).

This may be due to the fact that although the foreign population tends to maintain their taboos in host countries [43], they feel less social pressure than in their countries of origin, as is the case, for example, in a study conducted among sub-Saharan immigrants in France, where mention is made of different sexual socialisation depending on whether they emigrated to before or after the age of 10 [44], or, as previously mentioned, it may be because they directly identify the topic as a taboo subject that they simply do not want to talk about.

This tendency is not observed in the GRT group, who along with the Near–Middle East group, tend to discuss negative environmental perceptions of vaccination (B) more than people from other ethnicities/regions of origin. This may be because these are groups, particularly GRTs, who tend to give great importance to the extended family as a fundamental pillar of care and support [45]; for example, it is typical for the extended family to be present and care for their sick loved ones when they are treated in healthcare services [46]. This importance given to the extended family may explain the concern regarding the opinion of the environment. It is important to consider that the influence of region of origin on HPVV uptake is influenced not only by ethnicity and religion (discussed above) but also by language proficiency and years of residence in the country [7]. In this respect, some studies on the use of health resources (including preventive ones) by the immigrant population identify both language and a lack of knowledge of administrative requirements as barriers to access [47,48]; furthermore, the existence of a process of adaptation and acquisition of knowledge of the health system is noted, which leads to a lower use of these resources during the first years of stay in the country and to a similar level or even slightly higher level of use than the native population after some years in the country [49]. In terms of specific themes, taboos (A) are found to be present above all in conversations with people from Africa (NA; SSA) and in the Muslim population, which shows the interrelation between religion and ethnicity mentioned previously and in the literature [26]. However, while this trend detected by the HCPs is similar for the two regions of the African continent (although higher in the predominantly Muslim one), this is not mentioned for regions of the Asian continent, where the Western region (NOME), predominantly Muslim, stands out from the Eastern region (FE) in terms of the frequency of moral/cultural themes (A, B, C). It is worth noting, as Nevado Llopis points out, that in the communicative model of Asian cultures, it is common to affirm the doctor’s assertions without really communing with the information or having understood it, which can be misinterpreted by HCPs, assuming a “yes” and avoiding going deeper into these issues [6].

The Latin American population is the one in which the appearance of cultural/moral themes was least indicated. In the study carried out by Lechuga et al. [50] among the Latino population in the United States, they observed that the normalisation of sexuality was experienced in a very heterogeneous way by this group and had a great influence on the decision to vaccinate against HPV, and while, in part of this group, there is normalisation of sexuality and open communication between mothers and daughters on this topic, for others, it continues to be a taboo subject around which they feel great discomfort when talking. This normalisation–taboo polarisation could justify the fact that this topic is less common during discussions with this group, as it is not considered necessary by those who normalise it and it is avoided by those for whom it is taboo. Even so, it is important to emphasise that, as these authors point out, it is an indicative factor in the subsequent acceptance of vaccination. Difficulties in accessing the health system and completing the recommended schedules (G) were mentioned by the respondents as a theme that tends to emerge in vaccine recommendations most frequently in reference to the GRT population and the undocumented population. The results for the undocumented population are in line with studies by Bas-Sarmiento et al. and Lebano et al., which show that undocumented migrants have more complicated access to healthcare and identify barriers to access such as a lack of awareness of eligibility and administrative requirements; language difficulties; the use of false identification cards; fear of being reported and discriminatory practices or denial of care [47,48].

#### 4.1.3. Gender

Regarding the differences in the perception of HCPs on the themes that arise during the recommendation of vaccination between boys and girls, a greater predominance of informative themes is observed among boys. This is to be expected given that one of the themes is mainly oriented towards them (F) and is consistent with the literature indicating that parents of boys have more doubts about vaccination than parents of girls [51], so a greater demand for information could be logical.

In addition, the fact that the vaccine was introduced later for boys and was initially targeted at men who had sex with men or to protect a future partner may lead to a higher demand for information about the change in the recommendation and about the effects of HPV in boys, especially if we take into account that studies show that the motivation to collaborate to protect a potential future partner is often insufficient to convince parents/guardians to vaccinate their sons [52,53]. On the other hand, cultural/moral issues come up more frequently in discussions on recommending vaccination for girls, which is supported in the literature by pointing to the existence of a double standard, as sexual activity is more readily accepted in boys than in girls [54,55,56]. This tendency reflects structural values associated with gender that may be exacerbated depending on the environment [31]. Regarding the frequency of the theme of the benefit of vaccination in boys (F), it is important to point out that Newman et al. identified the perception of the benefits of vaccination and the recommendations of health personnel as the two factors that most influenced the decision to accept the vaccination of boys [57]. On the other hand, with respect to girls, although the frequency of the appearance of this theme in conversations is lower than in it is in conversations in which vaccination is recommended to boys themselves, it is still present, and this may be due to the protective factor that vaccination in boys may have on their future partners and the interest/doubt that this may generate in them.

#### 4.1.4. Level of Education and Language Proficiency

With regard to the levels of education and language proficiency, it is striking that the theme most frequently identified by the respondents in all groups is lack of information (E), except for the highly educated population, where the most common theme is misinformation (D). This may indicate a greater access to diverse sources of information by this group which allows them to identify differences between sources and generate doubts about the veracity of this information.

On the other hand, the theme of access (G) tends to be more present in conversations with the LPLL population than in the rest of the populations in this category. As explained above, language difficulties have been identified as a major barrier to accessing health services. Finally, the higher prevalence of a lack of knowledge about HPV and its consequences (E) among the LLE population and the LPLL population may be due to the complicated jargon that is often used by the health system and which is probably not understood, especially by these populations [58].

### 4.2. What Themes Emerge According to the Characteristics of the HCPs?

The results of our study show a tendency for younger HCPs to address more themes during the discussion than older HCPs. One possible explanation is that for some years now, there has been a growing trend and demand to include communication skills in the curriculum of undergraduate and postgraduate studies [59], which shows, on the one hand, the recognition on the part of healthcare personnel of the existing limitations in this field and, on the other hand, that educational systems are increasingly giving greater importance to these aspects and are including more activities related to this field. University degrees have undergone change with competency-based design [60], in which social, communicative and multicultural aspects have a greater place and importance, as shown by the fact that among the eight “Key Competences for Lifelong Learning” identified by the European Parliament and the Council of the European Union, communication in the mother tongue; communication in foreign languages; social and civic competences and cultural awareness and expression [61] are included. This would also partly explain how the younger population tends to be more prepared for and more sensitive to these issues.

Moreover, the dynamics of communication plays a key role in vaccine hesitancy and acceptance. In some cultures, discussing health-related topics such as vaccination can be sensitive, especially when sexual health is involved. In many cultures, there is a high respect for authority figures, often including HCPs, and among these, older professionals are often considered more authoritative. This respect may lead to less questioning and more acceptance of their recommendations without discussion [6], which may also help explain why the themes addressed decrease with the age of the HCPs. Younger professionals may not yet have the same level of authority and therefore receive more questions from patients.

In terms of professional profiles, paediatricians and school doctors are the ones with whom the most themes tend to arise, which may be partly explained by the fact that most children initiate HPV vaccination with these professionals, either with paediatricians [62] or with school doctors in the case of school vaccination strategies.

### 4.3. Considerations and Strategies Proposed by HCPs to Improve Communication about and Recommendation of HPV Vaccination

#### 4.3.1. Affordability

This is a theme that, at first sight, would escape the scope of the conversation in consultation, occupying higher levels if it were observed from a socio-ecological model in the case of a national (or regional) public policy [63].

Some of the recommendations given by the respondents are about the cost of the vaccine; even if this is perceived as an important barrier, the literature [64] suggests that when healthcare providers inform parents about the vaccine, even if it is not funded, they are more likely to accept the recommendation. On the other hand, it has been observed that several authors [65] recommend appealing to their moral responsibility to protect their children as part of the strategies to convince them of the convenience of vaccinating themes; we understand that this strategy should only be used with those people that healthcare providers consider can afford it or in systems where it is administered free of charge, as otherwise, it could generate a feeling of guilt in parents. On the other hand, a study by Reiter et al. found an association between willingness to pay for the vaccine and ethnicity and family income [66].

A review of the literature conducted by Alarcão and Zdravkova [18] concluded that there were women from disadvantaged backgrounds and a high concentration of ethnic minorities who refused papilloma vaccination; therefore, it is a double-edged issue—on the one hand, an economic issue, and on the other hand, linked with contextual themes for different ethnic minorities [67]. In our study, we considered that the affordability of the vaccine would be part of the difficulty in completing the recommended schedule (G). According to the HCPs surveyed, the GRT population is the group in which this theme emerges most during vaccine recommendation.

#### 4.3.2. Sexuality and Gender

This covers both cultural/moral and informational themes. Recommendations made by Cartmell et al. include, among the six HPV vaccination messages and communication strategies that can be instrumental in increasing uptake, a focus on cancer prevention rather than sexual transmission and emphasising the need for both boys and girls to be vaccinated [65]. The respondents indicated the importance of being sensitive in dealing with themes of sexuality and gender. As Marlow et al. point out, this may be especially important among immigrant populations due to the prevalence of cultural beliefs [68]. To this end, the respondents indicated that the focus can be shifted to the preventive aspect of the vaccine rather than sexual activity. In this regard, some authors point out that desexualisation techniques (shifting the focus of the message to cancer prevention rather than sexual activity) could be perceived as a means to facilitate vaccine uptake in certain population groups, such as those who mistakenly believe that HPV vaccination leads to increased promiscuity, those who believe that sexually transmitted infection (STI) protection is not relevant to their children or those who do not feel comfortable discussing their children’s sexuality. However, the authors point out that this strategy may not be appropriate for other groups, such as the children of parents who tend to be passive towards vaccination, parents who attach great importance to their role in sex education or adolescent boys with non-heterosexual orientations [69].

Regarding concerns about age and sexual activity, the respondents recommend indicating that the vaccine is not focused on their children’s current sexual activity but on protecting them in case they become sexually active in the future. In this regard, some authors such as Biancarelli et al. support moving the routine recommendation to 11 years of age (a practice that already occurs in some countries), arguing that at that age, parental concerns related to sexual activity are lower [70], while other authors such as Gordon et al. suggest that it is easier for parents to accept vaccination when their children reach a “more appropriate” age [19].

#### 4.3.3. Informative Materials and Campaigns

Regarding the use of visual aids in consultation, these have often been recommended in the literature [71], especially for populations with a low literacy level [72]. Such materials can be adapted to the needs of the population, so it is recommended to co-create them with representatives of the target population and to make them culturally specific [73].

Hafner et al. point out that in the context of medical consultation, visual aids can improve communication, especially in complicated situations such as when there are language barriers, aphasia, illiteracy or hypoacusis [74]. In addition, the literature points to the desirability of using the materials as a complement to and not as a substitute for oral interaction given that the materials independently may be unread, misinterpreted or misunderstood [24,74] and also to allow APGs to ask questions and exchange their concerns about the HPV vaccination with the HCPs [75].

Alberts et al. propose the strategy of using interaction-based approaches (face-to-face communication or social media) to enable parents/guardians to ask questions and share their concerns about HPV vaccination [75], which would be in line with the recommendation made by the respondents to conduct interactive workshops and training sessions for children/adolescents and/or their parents/guardians, where such interaction and the possibility to address questions and exchange views were sought. Additionally, the involvement of community leaders has proven to be beneficial in promoting effective communication and trust [76].

The use of mass media and social networks can have a major impact on vaccination intention, the generation of disaffection and recovery from vaccination. As discussed in our survey, the use of influencers close to the target population may be one approach to increasing vaccine uptake. Similar precedents exist for use in both vaccination campaigns [77,78,79] and other health interventions [80]. However, the use of social media remains an ambiguous decision and can generate both positive and negative results [81,82]. On the other hand, some intervention campaigns turned community members (e.g., pharmacists) into micro-influencers, showing favourable results [83].

In this type of campaign, we can differentiate between micro campaigns, using influencers, and macro campaigns, such as the Danish case. In Denmark, there was a sharp increase in vaccination refusal after negative media coverage, including the broadcast of the documentary *The Vaccinated Girls: Sick and Abandoned* and a subsequent rebuilding of trust through a national information campaign. The information campaign used social media to disseminate information, creating a website and Facebook page and promoting hashtags on Twitter, such as #stophpv. The campaign included information about HPV, its consequences and the vaccine and personal stories of women who had experienced cervical pre-cancer or cervical cancer. It also presented information specifically targeted at parents who had postponed vaccinating their children against HPV [84]. The Danish experience shows, in addition to the impact of the media and social media, the importance of early identification of such risks and responding quickly to them using social media and, if possible, a single spokesperson. However, the important role of HCPs should not be forgotten, and it is essential that they are well trained and informed, proactively recommend the vaccine and seek to clarify any concerns that APGs may have.

On the other hand, it should be noted that the literature points out that the use of social media and networks has a cultural component, that certain groups tend to use them differently than others and that the use of social networks is even noted as an indicator of integration of ethnic minorities [85]. For example, Taylor et al. noted that Cambodian women in the United States tended to be less informed by media sources than women from other racial/ethnic groups [86]. This is an aspect to consider when deciding which population to target and through which medium.

#### 4.3.4. Multilingualism/Multiculturalism

One of the recommendations made by the respondents on this category was to increase the use of translators or doctors who share the same language and cultural background as minorities. This aspect has already been pointed out in the literature [7]. This also implies using culturally sensitive language and communication styles [19,24,75] to ensure effective participation. In this sense, what is said and how it is said is as important as what is not said and why it is not said.

Our survey collected the recommendation of using translation apps as a means of overcoming the language barrier (but not the cultural barrier). The scientific literature has demonstrated the practical usefulness of these apps, despite their limitations [87], and their positive effect on the interaction between HCPs and interlocutors [88].

With regard to the suggestion to use interpretation services, previous studies indicate that although implementation is difficult, mainly due to financial and other resource constraints, when they were used, migrants were more likely to trust HCP diagnoses, and HCPs reported a clearer understanding of migrants’ symptoms [9].

The adaptation of materials should not be categorised in a way that accentuates prejudices but aimed towards personalisation of this particular concern which, as we have seen, seems to be present with different intensities among different groups and may even depend on other characteristics. The recommendations included in the Guidelines for Tailoring the Informed Consent Process in Clinical Studies [89] could be useful for adapting information materials to population groups.

#### 4.3.5. Quality of Care

One of the proposals for addressing different themes with different groups has been the use of motivational interviewing, which provides a space for questions and expressions of concern; this approach encourages open dialogue and helps to dispel any doubts or misconceptions [90]. There is no agreement in the literature on whether a persuasive approach is more or less effective [91]. However, CDC recommends assuming that parents will vaccinate and starting with a presumptive approach; if parents express concerns, making a strong recommendation for vaccination and if they remain hesitant, trying to understand their concerns and providing them with the requested information addressing their concerns [92]. The combined use of a presumptive approach and motivational interviewing plus an information sheet with undecided parents has been shown to be effective, but the communication skills of health professionals need to be improved for this to be undertaken efficiently [93].

Beyond recommendation itself, a lack of knowledge and trust in the health system can lead to a lack of access to it and/or a lack of trust in the vaccine; therefore, some strategies have been proposed to facilitate access, including some already mentioned in this article, such as the use of cultural mediators or translators, or, as Ghahari et al. indicate, developing programmes aimed at improving migrants’ knowledge, skills and trust in the health system [94]. In addition, as mentioned in the survey, it would be advisable to keep APGs linked to the same HCP, avoiding a change in professionals and thus facilitating the creation of a personal relationship that enables trust.

It is important to consider that for a correct approach to vaccination recommendation by HCPs, in addition to improving their communication and intercultural skills, knowledge about the infection and the vaccine must also be improved. Our study did not collect data on this knowledge from our respondents, but studies show that health professionals lack general HPV and vaccine knowledge and have low self-confidence in counselling and addressing parental concerns and discomfort in discussing sexual issues related to vaccination [95], so it would be advisable that strategies aimed at improving the training of HCPs include all of these areas.

#### 4.3.6. Collaboration with Schools

School-based vaccination has been shown to be effective in increasing vaccination rates, especially when combined with single-cohort vaccination [96]. Moreover, studies have shown that HPV vaccination rates are lower in schools with a higher density of ethnic minorities [29]. Therefore, it is important to design strategies based on the reality of the schools.

Among the proposals included in the answers of the survey, some were related to performing interventions in schools, such as including vaccination-related themes in the school curriculum or providing informative talks [97].

According to the literature, to promote vaccine uptake, future communication strategies should target not only parents but also close family members. Since family members have a significant influence on the decision-making process, educating the entire family about the benefits of the vaccine is important [14]. Combining curricular information with the intervention of school nurses can help to address the problems identified by school nurses and enhance their effectiveness [98].

In fact, the role of school nurses is highlighted in the literature, both in terms of their role as opinion leaders and trainers with parents, teachers and students and their impact in reducing health inequalities in the delivery of HPV programmes [99]. It is therefore very important that they are well trained and motivated.

Another collaboration with schools, although it does not appear in our survey but is indicated in the literature, is their role as a facilitating agent in catch-up campaigns for unvaccinated people; Gordon et al. point out re-offering the vaccine at a later age in faith schools can be an effective strategy to increase uptake among this population [19].

## 5. Limitations

As an exploratory study, its strength lies in its focus on a dimension that has been little explored at the European level, such as intercultural HCP–patient communication in relation to papillomavirus and its vaccine. However, there are a number of limitations inherent to the nature of the study, such as the lack of statistical validity of its results or of a random sampling method that could make it more representative of the whole population. Moreover, being an exploratory survey among HCPs, it only reflects the views of one of the actors in the communication. In an attempt to alleviate these limitations, a broad and thorough discussion of the trends observed was undertaken, and an effort was made during the discussion to include literature that incorporates the views of the APGs. Furthermore, the findings of this study may be a point of support for future research trying to determine possible inconsistencies or alignments between the perceptions of HCPs and the beliefs of the APGs themselves.

On the other hand, the tool used in the study makes it difficult to discern the influence of preconceptions or subjectivity on the responses; likewise, although it makes it possible to identify the themes that appear in conversations with APGs and to know whether they are observed by a higher or lower number of HCPs, it does not make it possible to know the frequency with which these themes appear. This is because it was decided to simplify the survey to facilitate a higher quality and number of responses. However, it should be kept in mind that this is only a guide given that differences between groups are merely trends identified by HCPs and not categorical differences; the intersectional component of the different categories and their interrelationship in each individual; that the specific circumstances of these populations in different contexts must be considered and that due to the exploratory nature of this study, the results cannot be extrapolated to the general European population.

## 6. Conclusions

Given the diversity of the population served and the risk of a misguided approach leading to the creation of pockets of unvaccinated populations, it is necessary for HCPs to adapt to the characteristics, concerns and interests of their audiences when recommending HPV vaccination.

The results of the study show how HCPs perceive different trends in information needs and interests related to HPV and the HPVV depending on the characteristics of the people they serve. Aspects such as a lack of knowledge about the infection or misinformation about the vaccine tend to be present in most populations.

Strategies should be explored that involve APGs, for example, by improving their awareness and knowledge of HPV infection and the vaccine or by funding the vaccine; and strategies involving HCPs, for example, by improving their training in multicultural communication and providing them with the necessary tools to facilitate the recommendation (consultation support materials, application and translation assistants, etc.) and even third parties, such as involving schools, cultural mediators or influencers.

Future research should continue to explore and develop effective strategies for communication in settings where HCPs and APGs do not share a common background, with the ultimate goal of improving healthcare outcomes for all people, including uptake of HPV vaccination.

## Figures and Tables

**Figure 1 vaccines-12-00748-f001:**
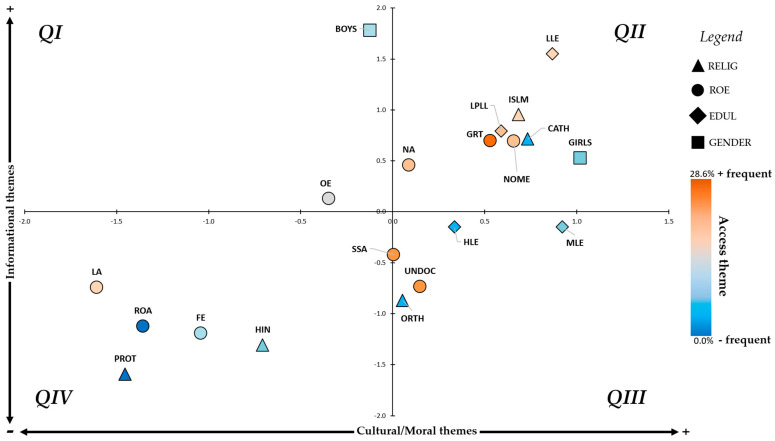
Scatter plot of the different populations according to the thematic categories that typically appear during HPV vaccination recommendation.

**Table 1 vaccines-12-00748-t001:** Characteristics of the sample.

		n (%)
Professional profile	Family doctor	12 (5.66%)
	Paediatrician	19 (8.88%)
	Nurse	110 (51.89%)
	School medicine	57 (26.64%)
	Other speciality	14 (6.54%)
Country of practice	Bulgaria	1 (0.49%)
	Estonia	1 (0.49%)
	Finland	1 (0.49%)
	Turkey	1 (0.49%)
	Austria	3 (1.46%)
	The Netherlands	10 (4.88%)
	Malta	11 (5.37%)
	Slovenia	13 (6.34%)
	Spain	69 (33.66%)
	Croatia	95 (46.34%)
Age group	≤35	46 (22.11%)
	36–45	60 (28.85%)
	46–55	53 (25.48%)
	≥56	49 (23.55%)
Gender	Man	14 (6.7%)
	Woman	195 (93.3%)
	Other	0 (0.0%)

Missed: Speciality = 2; Country = 3; Age group = 6; Gender = 5.

**Table 2 vaccines-12-00748-t002:** Respondents claiming to serve populations with each characteristic (multiple-response answers).

	Population	n (%)
Religion	PROT	18 (8.41%)
	ORTH	19 (8.88%)
	CATH	200 (93.46%)
	ISLM	90 (42.06%)
	HIN	26 (12.15%)
	BUD *	5 (2.34%)
	JUD *	6 (2.80%)
	Other *	9 (4.21%)
	IDS *	6 (2.80%)
Region of origin of immigrants and descendants of immigrants, ethnicity and/or legal status in the country	OE	129 (60.28%)
	NOME	69 (32.24)
	FE	52 (24.30%)
	NA	64 (29.91%)
	SSA	36 (16.82%)
	LA	66 (30.84%)
	ROA	15 (7.01%)
	GRT	49 (22.90%)
	UNDOC	37 (17.29%)
	IDS	50 (23.36%)
Gender (of the young adolescent eligible for vaccination)	Boys	195 (91.12%)
	Girls	203 (94.96%)
	Other *	10 (2.42%)
	IDS *	5 (1.21%)
Educational level and language proficiency	LLE	123 (57.48%)
	MLE	168 (78.50%)
	HLE	130 (60.75%)
	LPLL	68 (31.78%)
	IDS	17 (7.94%)

N = 214. Religion: Protestantism (PROT), Orthodoxy (ORTH), Catholicism (CATH), Islam (ISLM), Hinduism (HIN), Buddhism (BUD), Judaism (JUD), Other; Region of origin of immigrants and descendants of immigrants, ethnicity and/or legal status in the country: Other European (excluding the country’s population) (OE), the Near or Middle East (Turkey, Syria, Jordan, Egypt, Pakistan, Iran, etc.) (NOME), the Far East (China, Korea, Japan, Philippines, Thailand, etc.) (FE), North African (NA), Sub-Saharan Africa (SSA), Latin America (LA), the rest of America (ROA), Gypsies, Roma and Travellers (GRT), undocumented migrant population (UNDOC); Educational level and language proficiency: Low level of education (incomplete school education) (LLE), medium level of education (completed school education or vocational training) (MLE), high level of education (university or higher education) (HLE), low proficiency in the local language (LPLL). All: I do not serve any population with these characteristics (IDS). * Less than 5% of respondents report serving populations with this characteristic, so they have not been included in the presentation of the results nor in the discussion.

**Table 3 vaccines-12-00748-t003:** Proportion of respondents who consider that this theme comes up during discussions recommending HPV vaccination based on their age and speciality (multiple-response answers).

	A. Taboos	B. N-Percept	C. Lack Benef	D. Misinfo	E. Lack Known	F. MV	G. Access	None
**Age**	**% (n)**	**% (n)**	**% (n)**	**% (n)**	**% (n)**	**% (n)**	**% (n)**	**% (n)**
≤35	67.39 (31)	67.39 (31)	67.39 (31)	80.43 (37)	82.61 (38)	50 (23)	32.61 (15)	26.09 (12)
36–45	50 (30)	55 (33)	53.33 (32)	71.67 (43)	75 (45)	41.67 (25)	33.33 (20)	31.67 (19)
46–55	54.72 (29)	56.60 (30)	50.94 (27)	62.26 (33)	75.47 (40)	32.08 (17)	33.96 (18)	33.96 (18)
≥56	44.90 (22)	59.18 (29)	48.98 (24)	65.31 (32)	61.22 (30)	36.73 (18)	22.45 (11)	26.53 (13)
**Speciality**	**% (n)**	**% (n)**	**% (n)**	**% (n)**	**% (n)**	**% (n)**	**% (n)**	**% (n)**
Nurse	54.29 (57)	49.52 (52)	47.62 (50)	64.76 (68)	71.43 (75)	36.19 (38)	31.43 (33)	33.33 (35)
Fam. D	58.33 (7)	41.67 (5)	41.67 (5)	50 (6)	66.67 (8)	41.67 (5)	33.33 (4)	41.67 (5)
Paed.	62.5 (10)	68.75 (11)	75 (12)	68.75 (11)	81.25 (13)	62.5 (10)	37.5 (6)	25 (4)
School M	50 (28)	71.43 (40)	62.5 (35)	78.57 (44)	78.57 (44)	37.5 (21)	21.43 (12)	19.64 (11)
Other	33.33 (6)	55.56 (10)	33.33 (6)	61.11 (11)	55.56 (10)	33.33 (6)	33.33 (6)	11.11 (2)

Themes: A. Taboos related to people’s sexuality that may affect the conversation between a healthcare professional and an adolescent or his/her parents/guardians (A. Taboos); B. Negative perceptions of the HPVV in their environment (e.g., that it encourages promiscuity) and how this perception influences their decision to be vaccinated (or not) (B. N-percep); C. Considerations about the lack of benefit of the vaccine at the time, HPV being a sexually transmitted infection and given that it is administered at a very young age (C. Lack benef); D. Misinformation about the HPVV, such as lack of efficacy and safety or unproven adverse effects (e.g., that it causes infertility) (D. Misinfo); E. Lack of knowledge about HPV infection and its consequences (E. Lack known); F. When offering the vaccine to a male, the false belief that it only has health benefits for men who have sex with men (F. MV); G. Difficulties in accessing the health system and completing the recommended schedules (2 or 3 doses) (G. Access). Age: 35 years old or below (≤35; N = 46), between 36 and 45 years old (36–45, N = 60), between 46 and 55 years old (46–45, N = 53), 56 years old or above (≥56, N = 49); Speciality: Nurse (Nurse, N = 110), family doctor (Fam. D, N = 12); paediatrician (Paed., N = 19); school medicine (School M, N = 57), other speciality (Other, N = 14).

**Table 4 vaccines-12-00748-t004:** Strategies proposed by HCPs to improve communication and recommendation of HPV vaccination.

Theme	Codes	n ^a^	Theme Verbatim Examples ^b^
Affordability	Cost of vaccine and public funding	5	“The difficulty when explaining the HPV vaccine is the economic cost that it brought to the family. If the vaccine is offered at ages where it is financed by the national health system, there is no doubt”.
	Precarious situation	1	“Many immigrants have a precarious economics situation and cannot afford to buy the vaccine, although this can happen with any population group”.
Sexuality and gender	Appropriateness of the vaccine for both sexes	10	“It is important to emphasize that it is not exclusively a female disease; all possible types of malignant changes are shown. Explain to them the possibility of spreading the infection, but also emphasize that there is no screening for young men”.
	Concern about the age of vaccination	3	“Some parents think that we can wait to vaccinate because their children are too young to have sexual relations”.
	Debunking false beliefs (cause of promiscuity and infertility; form of contagion)	5	“Explain that HPV can be contracted even by people who have had one sexual partner”.
	Communication strategies for dealing with taboo topics	8	“When the practitioner is going to talk about sex and the illnesses around, it may be better to ask the parents or the guardian to leave the room for a while; the adolescents are usually shy by the presence of their tutors and they can be too ashamed to ask the questions they have. Then, to inform about the vaccine first the teenager and then the guardian so even if the guardian is close-minded about the vaccine, the young adult can have the information to form their own opinion and come back when they don’t need the guardian’s permission”.
Informative materials and campaigns	Supporting materials to the consultation	1	“Communication and discussions can be recognized by various means other than verbally, for example, using brochures, short media films, and anything that goes well within the less education”.
	Workshops and campaigns	3	“Workshops and training sessions for the entire susceptible population”.
	Opinion leaders/social networks	4	“Our children are likely to listen some idiot on TikTok, Instagram, etc., than MD. Perhaps we can use such people to say something good about HPV vaccination on TikTok, Instagram, etc.”
Multilingualism	Multilingual information	5	“If they don’t understand the language, the recommendation is very difficult; there is a lack of supporting material”.
	Translation apps	1	“Google translator, both written and the version to listen to.”
	Translator/cultural mediator	3	“Having a translator to communicate with the parent other than the child”.
Quality of care	Build trust	3	“It’s better if they come to the same medical practitioner every time, because if they have to explain their situation every time they come to the centre, they will be fed up and simply won’t come again”.
	Use conversational techniques, such as motivational interviewing	4	“Parents should be well informed and educated about HPV and reassured about vaccination effects. Usually, it’s enough to talk to parents and have time for their questions and insecurities”.
	Catch-up strategy at older ages	1	“When they go to the adult clinic, they should insist on vaccination in the same way as in the paediatric clinic with external vaccinations”.
	Improve communication skills and awareness of the HCP	1	“Encourage vaccination through conversation and avoid it acting as a biased persuasion from which the doctor/nurse benefits financially or otherwise”.
Collaboration with the school	Inclusion in school programme	2	“Vaccine-preventable diseases and vaccination topics have to be included into the school programme for the middle educational level in Estonian and Russian”.
	Communication and discussion in school (students, parents/guardians, teachers)	5	“Awareness-raising and training campaigns/lectures in high schools targeting parents on the one hand and adolescent boys on the other hand”.
	School vaccination program	1	“I inform them that if they wait after 14 years old, they have to vaccinate 3 times instead of 2, and normally, they accept to vaccinate at school at the normal age…”
	Addressing the different approaches	1	“In Catholic religion, we have problems in sharing wrong information about HPV vaccine in school by teacher of biology and teacher of religion. Also, we have problem with parents who share their opinions through social network”.

^a^ Some verbatim responses could be assigned to more than one code. ^b^ Some verbatim responses were not originally in English and have been translated for the article.

## Data Availability

The original data presented in the study are openly available in Zenodo at https://zenodo.org/records/11276085. Accessed on 5 June 2024.

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
