# Peer review of "Healthcare Professionals’ Perspectives on HPV Recommendations: Themes of Interest to Different Population Groups and Strategies for Approaching Them"

_vaccines, 2024, doi:10.3390/vaccines12070748_

Round 1
Reviewer 1 Report
Comments and Suggestions for Authors
Thanks for the opportunity to review this article. I have some comments as follow:
1. Why cultural background or culture is not included which according to previous studies have more impact on the vaccination coverage than any other factor?
2. The materials and method section is not well detailed How this survey was conducted, sampling strategy, rejection level of participation, why some religions were included ten fold higher in number than others....In this case are they the representative of their specific group of people?
3. The conclusion needs to be crispy and sexy. The current conclusion is lengthier than abstract, its better to shorten it, so its more attractive and acceptable to readers.
Author Response
Thank you very much for taking the time to review this manuscript. Please find the detailed responses below and the corresponding revisions/corrections in track changes in the re-submitted files. |
Point-by-point response to Comments and Suggestions for Authors |
Comments 1: Why cultural background or culture is not included which according to previous studies have more impact on the vaccination coverage than any other factor? |
Response 1: We have decided not to include the concept of culture directly because of the complexity involved in defining and measuring it precisely. Culture encompasses a wide range of behaviours, beliefs and values that are difficult to quantify and can vary significantly even within seemingly homogenous groups. Instead, we have preferred to include more specific, identifiable and measurable culture-related factors such as religion, region of origin and ethnicity, as we believe these factors provide valuable and more directly usable information for understanding attitudes and behaviours related to vaccination. In addition, these factors are also commonly reported in the literature, which facilitates discussion. |
Comments 2: The materials and method section is not well detailed How this survey was conducted, sampling strategy, rejection level of participation, why some religions were included ten fold higher in number than others....In this case are they the representative of their specific group of people? |
Response 2: Thank you for pointing this out. Agree. We have improved the description and structure of the methodology to make it more complete and clearer. Even so, it is important to note, on the one hand, that this is an exploratory study and, on the other, that it is the HCPs themselves who have indicated the populations they serve, so that the disparity in the representation of religions is due to the frequency and accessibility of these populations in the areas surveyed or simply to their being more (or less) identifiable. Furthermore, given the inherent limitations of an exploratory study, we have tried to be especially cautious in the discussion with the populations underrepresented in our survey to avoid extrapolating conclusions to the general population, always trying to rely on the existing literature.    |
Comments 3: The conclusion needs to be crispy and sexy. The current conclusion is lengthier than abstract, its better to shorten it, so its more attractive and acceptable to readers. |
Response 3: Thank you for pointing this out. We have shortened and improved the wording of the conclusion. |
Reviewer 2 Report
Comments and Suggestions for Authors
The work submitted for publications investigates an interesting and current question in the field of healthcare.
The manuscript flows well and is structured in a logical and appropriate manner.
The reviewer would like to offer a few points for consideration by the authors listed below:
1. Consider providing more information on the inclusion and exclusion criteria of participants in the study.
2. Consider providing the rationale for the sample size.
3. Also consider the use of terms "sex" and "gender". When referring to the biological sex then the appropriate term to use is "sex" while the use of the term "gender" refer to the sexual identity of a person.
good work overall.
Comments on the Quality of English LanguageEnglish language is OK.
Author Response
Thank you very much for taking the time to review this manuscript and recognising the interest of the topic in the current field of health care. Please find the detailed responses below and the corresponding revisions/corrections in track changes in the re-submitted files. |
Point-by-point response to Comments and Suggestions for Authors |
Comment 1: Consider providing more information on the inclusion and exclusion criteria of participants in the study. |
Response 1: Thank you for pointing this out. Agree. We have improved the wording and structure of the methodology and included this information. |
Comment 2: Consider providing the rationale for the sample size. |
Response 2: No sample size was set during the research design, as this is an exploratory study and the focus is more on the breadth and variety of responses than on statistical representativeness. The aim of this study is to identify trends and gain an initial understanding of the issues of interest regarding HPV vaccination in different population groups in Europe, rather than to provide statistically representative results on a large scale. Therefore, due to the design characteristics of this research, information on sample size was not established. |
Comment 3: Also consider the use of terms "sex" and "gender". When referring to the biological sex then the appropriate term to use is "sex" while the use of the term "gender" refer to the sexual identity of a person. |
Response 3: Thank you very much for your indications, we have reviewed and modified the text to use both terms correctly and consistently. However, if there is any usage with which you do not agree, please let us know. |
Reviewer 3 Report
Comments and Suggestions for Authors
Dear Authors,
Below I present my assessment of the manuscript:
Characteristic of the sample:
In a multi-center and international study, only one HCP responded from several countries? In principle, only the opinions of physicians from Spain and Croatia should be taken into account.
Table 4 – the notation n/N is unnecessary; % is enough; however, it is worth adding what N was? Could multiple answers be selected here? This is the only way to explain that the values ​​do not add up to 100%??
But this needs to be explained.
Lines 168-171 – a description for Table 5 is needed; a table without a description is very illegible; in addition, Table 5 – the abbreviations used in the table require explanation
Unfortunately, the population metric variables are very weak, but it is possible to calculate the approach to the subject of HCPs of different ages. This would be interesting, because the success of preventive programs does not depend only on the patients, but also on those who offer them such a program; and the method of conducting the program can be adjusted to the differences in the views of educators, differences that in this case would perhaps depend on age.
Chapter 3.4 should be supported by some specific numerical data; it is not known whether this is a further research part of the work or already a discussion?
The authors have undoubtedly taken up a very interesting and current topic – both due to the need to promote vaccinations and due to the migration of people from various parts of the world.
However, the overall impression is not high, the text is difficult to understand, and requires significant improvement. In principle, only table 5 and figure 1 are the result of research work; I propose to shorten chapter 3.4 and replace it with clear numerical data. This text is too long and difficult to read.
Comments on the Quality of English LanguageThe text needs to be checked.
Author Response
Thank you very much for taking the time to review this manuscript. Please find the detailed responses below and the corresponding revisions/corrections in track changes in the re-submitted files.
|
Point-by-point response to Comments and Suggestions for Authors |
Comment 1: In a multi-center and international study, only one HCP responded from several countries? In principle, only the opinions of physicians from Spain and Croatia should be taken into account. |
Response 1: Thank you for your comment and for highlighting the lack of clarity in our article on this point. Firstly, each HCP are giving their perspective about the population they serve, not about several countries. Secondly, the article is about an exploratory study to identify differences in the interests and needs of the population regarding HPV vaccination according to their characteristics and to identify strategies to improve the recommendation of HPV vaccination. As this is an exploratory study, the aim is not to obtain a sample size and statistical significance but to obtain a broad view of a phenomenon that has not been widely studied, at least not in Europe and not taking into account so many different dimensions. Last but not least, we do not see the distribution of the sample as problematic, given that we do not intend the data to be extrapolated to a specific country, but rather to allow us to understand the differences between an increasingly diverse population and between population groups that in many cases, regardless of where they are located, have common trends. Therefore, although the description of the sample shows that there is a greater predominance of responses from Croatia and Spain, as well as from women and nurses, we do not believe that it is necessary to eliminate the rest of the responses that allow us to increase our knowledge of the populations attended, since, as explained, we do not want to extrapolate the results to any of the countries. We are simply describing where our responses come from so that the reader can situate themselves, even so, and given that we provide access to both the survey and the database, we understand that those who are interested in knowing the responses from only one country can do so easily, since due to the exploratory purpose of the study, the analysis is based on descriptive statistics and does not seek complex relationships between the variables. All relevant explanations have been included in the article. We hope that the new description will make it more comprehensible to the reader, as it is certainly very important that the exploratory nature of our study is well understood. Thank you again |
Comment 2: Table 4 – the notation n/N is unnecessary; % is enough; however, it is worth adding what N was? Could multiple answers be selected here? This is the only way to explain that the values do not add up to 100%?? But this needs to be explained. |
Response 2: Thank you for your comment. Agree. Table 4 (now 2) has been modified. We have added in the title of tables 4 and 5 (now 2 and 3) “(multiple response answers)” to clarify it. Now, table 4 and 5 (2 and 3) include the N in the legend |
Comment 3: Lines 168-171 – a description for Table 5 is needed; a table without a description is very illegible; in addition, Table 5 – the abbreviations used in the table require explanation |
Response 3: The description in table 5 has been improved. A legend has been added to explain the abbreviations used and the N for each category. |
Comment 4: Unfortunately, the population metric variables are very weak, but it is possible to calculate the approach to the subject of HCPs of different ages. This would be interesting, because the success of preventive programs does not depend only on the patients, but also on those who offer them such a program; and the method of conducting the program can be adjusted to the differences in the views of educators, differences that in this case would perhaps depend on age. |
Response 4: Thank you for noticing this, it is a valuable input. A new section has been added in both results and discussion to show the differences in the topics covered by respondents according to their age and professional speciality. |
Comment 5: Chapter 3.4 should be supported by some specific numerical data; it is not known whether this is a further research part of the work or already a discussion? |
Response 5: Agree. Chapter 3.4 has been modified. The text has been replaced by a table with numerical data, the coding of the categories and examples of the literal wording of the responses. This has made it shorter and hopefully more understandable. |
Comment 6: The authors have undoubtedly taken up a very interesting and current topic – both due to the need to promote vaccinations and due to the migration of people from various parts of the world. However, the overall impression is not high, the text is difficult to understand, and requires significant improvement. In principle, only table 5 and figure 1 are the result of research work; I propose to shorten chapter 3.4 and replace it with clear numerical data. This text is too long and difficult to read. |
Response 6: Thank you to recognising the interest of the topic in the current field of health care. Tables 1 and 2 have been deleted, several texts and sections have been modified and section 3.4 has been completely modified according to the reviewer's indications. We hope to meet the reviewer's expectations. |
Round 2
Reviewer 2 Report
Comments and Suggestions for Authors
The authors have made a reasonable effort to address reviewer's comments.
Reviewer 3 Report
Comments and Suggestions for Authors
Thank you for improving your manuscript.
Comments on the Quality of English LanguageI think minor editing of English language required